# Role of Nanomaterials in COVID-19 Prevention, Diagnostics, Therapeutics, and Vaccine Development

Unnati Patel * ，Kavini Rathnayake ，Emily C. Hunt and Nirupama Singh

Department of Chemistry, The University of Alabama in Huntsville, Huntsville, AL 35899, USA
* Correspondence: up0004@uah.edu

**Abstract:** Facing the deadly pandemic caused by the SARS-CoV-2 virus all over the globe, it is crucial to devote efforts to fighting and preventing this infectious virus. Nanomaterials have gained much attention after the approval of lipid nanoparticle-based COVID-19 vaccines by the United States Food and Drug Administration (USFDA). In light of increasing demands for utilizing nanomaterials in the management of COVID-19, this comprehensive review focuses on the role of nanomaterials in the prevention, diagnostics, therapeutics, and vaccine development of COVID-19. First, we highlight the variety of nanomaterials usage in the prevention of COVID-19. We discuss the advantages of nanomaterials as well as their uses in the production of diagnostic tools and treatment methods. Finally, we review the role of nanomaterials in COVID-19 vaccine development. This review offers direction for creating products based on nanomaterials to combat COVID-19.

**Keywords:** COVID-19; prevention; diagnostics; therapeutics; vaccine





## 1. Introduction

According to the World Health Organization, since the beginning of the pandemic in 2019, over 535,863,950 confirmed infections and 6,314,972 deaths caused by the coronavirus disease-19 (COVID-19) have been reported worldwide [1]. COVID-19 presents a set of severe respiratory symptoms and has been shown to be transmissible through aerosolized droplets. COVID-19 is caused by severe acute respiratory syndrome coronavirus 2 (SARS-CoV-2), a member of a family of enveloped single-stranded positive sense RNA viruses called coronaviruses, which include severe acute respiratory syndrome (SARS) and Middle East respiratory syndrome (MERS) [2]. Like other coronaviruses, SARS-CoV-2 is surrounded by a lipid membrane containing several important proteins involved in the assembly and infection of the viral capsid [3].

The viral capsids of coronaviruses contain several proteins, including the spike glycoprotein. The spike glycoprotein is essential in viral recognition, binding, and entry to host cells [4]. The spike glycoprotein contains two binding subunits, S1 and S2. The S1 subunit can bind directly to host cell angiotensin-converting enzyme 2 (ACE2) receptors, which allow the virus to be taken up via clathrin-mediated endocytosis [5]. Acidification of the endosome causes conformational changes in the S2 subunit, which promotes membrane fusion, releasing the viral genome into the host cell [6]. Although this mechanism of coronavirus infection is well studied [7,8], evidence for alternate mechanisms exists. For example, clathrin-independent endocytosis methods have also been observed, indicating that the viral infection may be mediated through a variety of pathways depending on host cell type [7,9]. Furthermore, studies have shown that under certain conditions, viral infection is not pH-dependent and may be mediated by host cell proteases [8–11]. However, receptor-mediated endocytosis is considered the primary mechanism for SARS-CoV-2 infection.

Nanomaterials have been shown to be useful in the prevention, diagnosis, and treatment of viral infections [12]. Nanomaterials for the prevention of virus transmission

through disinfection of surfaces, such as protective equipment, have primarily focused on materials with antimicrobial or self-sterilizing properties. Inherently antimicrobial materials, such as silver and copper, have been shown to be effective against SARS-CoV-2 [13–15]. Other nanomaterials have physical properties that make them ideal for coating objects to allow for filtering of viral particles and self-sterilization [16–20]. These nanomaterials have been shown to be useful for coating personal protective equipment to enhance its effectiveness in preventing viral transmission [21–23].

Several nanoparticle-based diagnostic assays for viruses have been developed, which are typically based on receptor-ligand interactions [24]. These biosensors commonly use electrochemical [25–28] or optical [29–32] detection methods. Due to their good sensitivity and affordability, nanoparticle-based assays have been suggested for the detection of SARS-CoV-2 [33]. Several nanosystems have been shown to be effective in detecting SARS-CoV-2 viral RNA and antibodies [34–37]. These results indicate that nanomaterials can provide rapid, cost-effective, and accurate diagnostics for COVID-19, allowing for faster treatment.

Nanoparticles have been suggested as potential therapeutics for COVID-19 [38,39]. Nanomaterials have been shown to be effective antiviral treatments for both their inherently antimicrobial properties and their ability to serve as carriers for antiviral drugs [40,41]. Furthermore, a major benefit of nanoparticle-based antiviral treatments is that nanomaterials can be conjugated with proteins, allowing for targeted interactions, thus allowing for direct delivery of antiviral agents to host cells, and reducing off-target effects. This mechanism shows promising results in the treatment of COVID-19 [42–45]. However, further work is still needed in the development of nanomaterial-based therapies for COVID-19.

One of the most important advances in combating COVID-19 has been the development of vaccines. Many vaccines were developed using traditional approaches, but both the BioNTech/Pfizer and Moderna vaccines, which were given USFDA approval, were designed to deliver viral mRNA using a lipid nanocarrier. Lipid nanoparticles (LNPs) protect RNA from degradation and deliver it directly to the cytoplasm, generating a robust immune response [46]. These vaccines were shown to be highly effective against SARS-CoV-2 [47,48], giving hope not only that vaccination could stop the spread of COVID-19 but also that nanotechnology-based vaccines could be effective against other human pathogens.

The last few years have seen rapid developments in nanotechnology for the prevention, diagnosis, treatment, and vaccination for COVID-19, but the evolution of new variants continues to necessitate research into effective therapeutic and diagnostic strategies. In this review, we will discuss the role nanomaterials have had in the development of approaches for prevention, diagnostics, therapeutics, and vaccination for COVID-19.

## 2. Role of Nanomaterials in the Disinfection of SARS-CoV-2

Early in the COVID-19 pandemic, prior to the development of vaccines and effective antiviral treatments, it became important to have methods of preventing viral transmission through sterilization. Disinfection of protective clothing, equipment, and surfaces often requires time and resources that were in short supply during the height of the pandemic. Thus, antimicrobial nanomaterials could provide a pathway to alternate methods to the typical disinfection protocols, with additional clinical benefits, such as self-disinfecting properties, as well as economic advantages [49,50]. Several broad-spectrum antimicrobial nanomaterials are known, although none have achieved widespread use. These nanomaterials have provided strategies for the development of protective equipment and disinfectants aimed at preventing the transmission of SARS-CoV-2.

Some inherently antimicrobial materials have long been recognized, and their incorporation into nanomaterials has increased their effectiveness, due to the gradual release mechanism of toxic ions from the nanomaterial. Recent work has shown that AgNPs have effective antiviral activity against SARS-CoV-2, due to their ability to inhibit viral entry to cells [13,14]. Furthermore, sputter coating of AgNPs on surfaces, such as masks, to has been shown to prevent SARS-CoV-2 infectivity [13].

Another material with a long history of antibacterial activity is copper. Previous research has demonstrated that copper-based nanomaterials are efficient against a variety of viruses [51], and other coronaviruses, such as SARS [52]. Viruses exposed to copper surfaces show damage to the viral genome and morphological changes, such as deterioration of the viral envelope, indicating that copper nanomaterials may irreversibly inactivate viruses through a variety of mechanisms [53]. Additionally, masks infused with copper oxide have been shown to have antiviral properties when tested against two influenza viruses, H1N1 and H9N2, without affecting the physical properties of the mask [54], indicating that copper can be incorporated into textiles and maintain its antiviral properties, which would be important for creating clothing and protective equipment to prevent transmission of SARS-CoV-2. A recent study by van Doremalen, et al. showed that the titer of SARS-CoV-2 on copper surfaces is quickly reduced, with no viable SARS-CoV-2 detected on copper after 4 h [15]. These results offered further suggestion that copper-based nanomaterials could provide a useful strategy for SARS-CoV-2 prevention.

Titanium dioxide (titania, $TiO_2$) has also been suggested as a nanomaterial with potential antiviral activity against SARS-CoV-2. Although its antimicrobial activity has only been known since 1985, when its photocatalytic properties were first exploited for sterilization [55], irradiation of $TiO_2$ with UV light has since proven to be effective against a wide variety of microorganisms, due to its production of reactive oxygen species, which cause cell lysis [16]. $TiO_2$ has been shown to be effective against many human viruses [56–59], including SARS [60], which indicates that it may have antiviral activity against SARS-CoV-2. Recent findings show that $TiO_2$, induced by UV radiation, was effective on HCoV-NL63, a close genetic relative of SARS-CoV-2 [61]. These results provide further indication that $TiO_2$ nanoparticles could be an effective strategy to prevent SARS-CoV-2 infection. The photocatalytic properties of $TiO_2$ are particularly of interest in the prevention of SARS-CoV-2 transmission due to their usefulness in sterilization and in creating hybrid strategies for antiviral activity.

Recently, other nanomaterials have also attracted interest for their promising antimicrobial properties as well as their physical properties, which allow for self-sterilization and hybrid approaches to the prevention of viral transmission. For example, graphene (G) and graphene oxide (GO) have photothermal and superhydrophobic properties, which make them ideal for coating surfaces to allow for heat sterilization and to repel droplets from aerosols [22,62]. The single-layer negatively charged structure of G and GO contribute to their antiviral properties [17,20]. Additionally, these nanomaterials can be modified with functional groups to increase their electrostatic and hydrophobic interactions [20] or with other nanomaterials, such as AgNPs to form nanocomposites which increase their antimicrobial effect [63,64]. G, GO, and their derivatives have been shown to be effective in inhibiting the infection of a number of viruses [17,20,63–67], including the feline coronavirus [63].

For these reasons, it has been hypothesized that G, GO, and their derivatives might prove effective in preventing SARS-CoV-2 infection and transmission [62,68]. G and GO had previously been shown to make effective and breathable air filters [69,70], making them natural choices for the coating of facial masks. A recent study has shown that G can be used to coat commercially available disposable surgical masks, making them reusable, self-sterilizable, and recyclable, as well as imbuing the masks with the superhydrophobic and photothermal properties of G [22]. Using a dual-mode laser-induced forward transfer to additively deposit superhydrophobic G onto surgical masks, Zhong, et al. were able to create a porous coating with antimicrobial, photothermal, and superhydrophobic properties that was compatible with current methods of surgical mask production [22]. The superhydrophobic properties of the mask coating allow water droplets to roll off the mask surface [22], which is important in the prevention of SARS-CoV-2 spread, since the virus is predominantly spread through aerosols. The photothermal properties of the G coating also allows the surface temperature of the mask to reach 80 °C in sunlight, which is a high enough temperature for sterilization of viruses [22]. Finally, the masks are recyclable for

solar desalination, with steam generation rates greater than 1.13 kg/m$^2$ per hour and better salt rejection when compared to other membranes, due to their microporous structure [22]. These results indicate that G-coated masks might provide an environmentally sustainable method of preventing viral transmission. This antiviral activity may extend to SARS-CoV-2, as recent findings have indicated that G and GO functionalized face masks can trap and inhibit the infectivity of SARS-CoV-2 [23].

Polymeric nanomaterials have also been suggested for applications in preventing the spread of SARS-CoV-2. Polymers present many advantages as nanomaterials. Due to the wide variety of molecules that can be polymerized, the properties of polymers can be easily tailored to a specific application. Polymers without inherently antimicrobial activity have been used to create filters, which prevent particles of a certain size, such as aerosol droplets, from passing through the filter due to the small size of the pores in the material [19,21]. This has made polymeric nanomaterials ideal for producing or modifying facial masks. For example, polyvinylidene fluoride was used to create a multilayer mask capable of filtering simulated aerosolized SARS-CoV-2 particles [19]. Similarly, El-Atab, et al. created nanoporous membranes from polyimides which could be attached to an N95 mask and replaced for each use, allowing the mask to be used multiple times [21]. Pores ranging from 5 to 55 nm were achieved, allowing the modified mask to successfully trap the SARS-CoV-2 virus, which has a particle size of approximately 65 to 125 nm [21]. After using KOH etching on a silicon-on-insulator wafer to make a template for the porous membrane, the template is applied to a polyimide-coated silicon wafer [21]. Reactive ion etching was then used to transfer the template pores to the polyimide layer. The template could then be removed, and the polyimide membrane could be peeled off the silicon layer and attached to an N95 mask [21]. Furthermore, calculations of the airflow rate indicated that the porous membranes would be breathable, and the hydrophobicity of the polymer would contribute to the filtering ability of the masks, by repelling droplets which could contain SARS-CoV-2 viral particles [21]. To prevent the transmission of SARS-CoV-2, polymers with antiviral properties have also been suggested.

Several polymers have been previously shown to exhibit broad-spectrum antiviral activity [18,71–73]. Some of these polymers generate reactive oxygen species, which leads to their antiviral activity [71,72]. Other polymers have been shown to effectively block viral entry into cells by mimicking heparan sulfate proteoglycans, which are commonly targeted for viral attachment [18,73]. Recently, a similar mechanism of competitive binding has been used with nanosponges, containing a polymer core surrounded by the membrane of human lung epithelial cells, to prevent SARS-CoV-2 infections [74]. The biocompatibility of these polymers also makes them potentially well suited to the treatment and prevention of COVID-19 and other viral infections in vivo.

## 3. Role of Nanomaterials in Diagnostics of SARS-CoV-2 (COVID-19)

Diagnostics is a critical step in the fight against the COVID-19 pandemic to limit the spread of the disease as much as possible. Hence, several nanotechnology-based approaches have been developed for the detection of SARS-CoV-2, as summarized in Figure 1 [75]. Among a wide variety of SARS-CoV-2 detection methods available, the reverse transcription polymerase chain reaction (RT-PCR) takes most priority due to its specificity, sensitivity and simplicity in early detection of the viral genome [76,77]. RT-PCR for SARS-CoV-2 detection is based on the determination of coronavirus RNA levels, which appear after 5–7 days of infection, by using respiratory secretions [78,79].

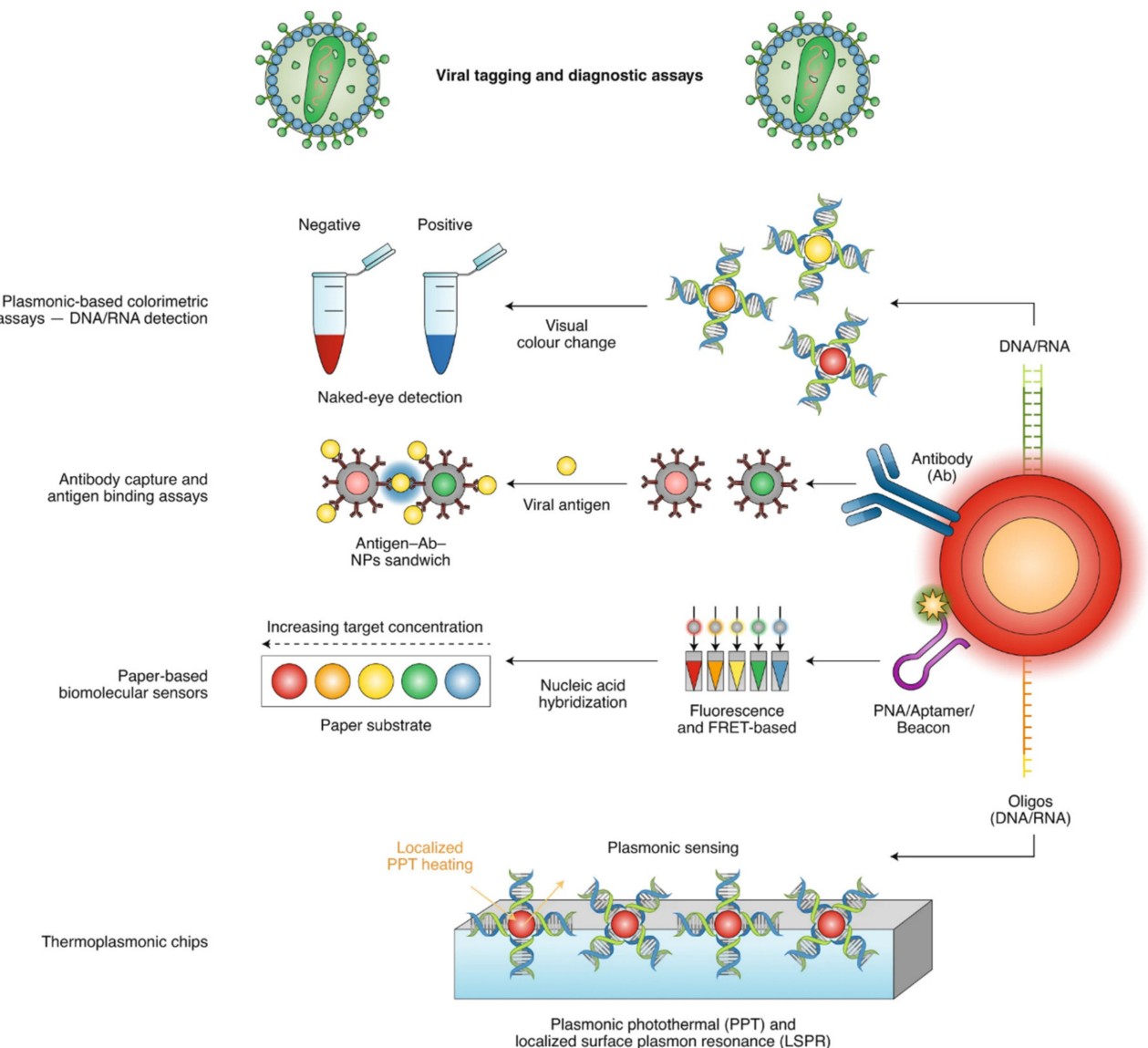

**Figure 1.** Nanotechnology-based detection approach for COVID-19. Adapted with permission from ref. [75]. Copyright 2020, Springer Nature.

Although PCR is the primary means of diagnosis of SARS-CoV-2, there are some drawbacks which have to be considered, such as the time used in the preparation of viral RNA, which affects diagnostic accuracy and leads to false positives in the results. Hence, there are other serological tests used such as enzyme-linked immunosorbent assay [80], chemiluminescence assay, immunofluorescence assay, or immunochromatographic test (ICT) for accurate SARS-CoV-2 diagnosis [81,82].

When developing new methods of detection, nanomaterials play a huge role due to their high surface area and ultra-small size [39,83]. There are examples of using nanomaterials not only in RT-PCR methods but also other virus detection methods, such as an enzyme-linked immunosorbent assay (ELISA) and reverse transcription loop-mediated isothermal amplification (RT-LAMP) [82,84,85]. One of the main SARS-CoV-2 diagnostic techniques is the lateral flow immunoassays or immunochromatographic test (ICT) which detects the antigens or antibodies. This point-of-care test method of detection, using a small amount of patient blood (10–20 µL), is a rapid, simple, and easy-to-use method without any laboratory facility or skilled specialists.

Metal, magnetic nanoparticles and metal-organic frameworks have been used in virus detection [86,87]. Gold nanoparticles (AuNPs) have been widely used in these types of viral detection methods due to their unique optical properties, which make them ideal candidates for rapid colorimetric diagnostic tests [88,89]. In this lateral flow immunoassay, the AuNPs are labeled with a surface antigen, which can specifically bind to SARS-CoV-2 antibodies (including both IgM and IgG), and are attached on the conjugation pad. Thereafter, the blood or serum sample is released into the sample pad then the antibodies start interacting with the antigen-bound AuNPs as the sample flows through the chromatographic lateral flow. These AuNP conjugates are dragged through the chromatographic strip by capillary action, resulting in corresponding lines. Hence, there will be two different lines: M (IgM) line for the anti-SARS-CoV-2 IgM antibodies and G (IgG) line for anti-COVID-19 IgG antibodies. The samples which do not have both antibodies will not bind to any labeled lines and will not show any lines. The remaining colloidal gold travels up the nitrocellulose to the control line zone, which captures the excess conjugate demonstrating that the fluid has migrated adequately through the device. This approach, which was presented by Li and coworkers, showed that the ICT-based detection method can detect both antibodies at the same time within 15 min with a sensitivity of 88.7% and specificity of 90.6 tested by using blood samples from PCR-confirmed COVID-19 patients and negative patients from different clinical sites (Figure 2) [90]. Ray group has reported anti-spike antibody-attached gold nanoparticles for the rapid diagnosis of specific COVID-19 viral antigen or virus via a simple colorimetric change observation within a 5 min time period [91]. However, the majority of current detection techniques collect patient samples through nasal or throat swabs. Pan and coworkers have followed a dual-prong approach, integrating nucleic acid amplification and plasmonic sensing using antisense oligonucleotides-capped plasmonic gold nanoparticles as a colorimetric reporter [92].

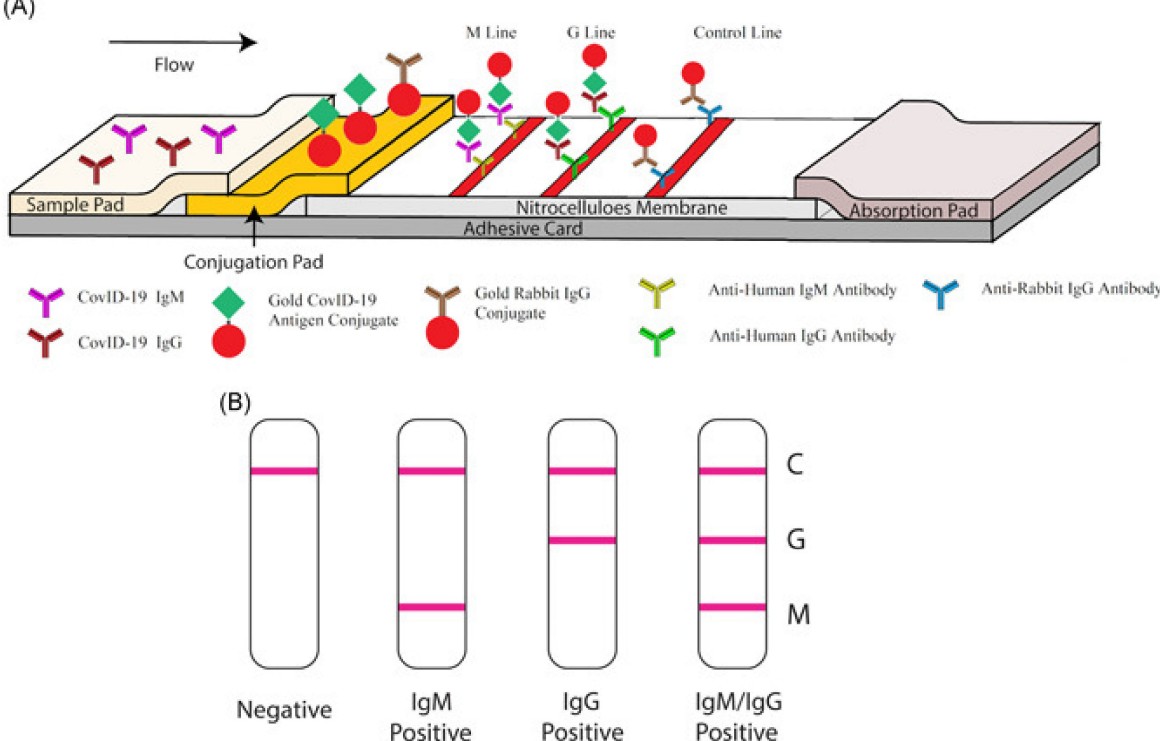

**Figure 2.** Schematic illustration of rapid SARS-CoV-2 IgM-IgG combined antibody test. (**A**), Schematic diagram of the detection device; (**B**), an illustration of different testing results, where C means control line; G means IgG line; M means IgM line. IgG means immunoglobulin G; IgM means im-munoglobulin M; SARS-CoV-2 means severe acute respiratory syndrome coronavirus 2. Adapted with permission from Ref. [90]. Copyright 2020, Wiley Periodicals, Inc.

Recently, carbon-based nanomaterials, for example, graphene, carbon nanotubes (CNTs), and carbon dots (CDs), have also gained attention in the development of diagnostics for COVID-19. These nanomaterials have photoluminescence, biocompatibility, and high stability, which distinguish them for applications, including biosensing and bioimaging [93]. The use of graphene nanoparticles for diagnostics of respiratory viruses, including SARS-CoV-2, has been reported [94]. Recently Seo and co-workers have reported the use of a field-effect transistor (FET)-based biosensor platform for rapid SARS-CoV-2 detection (Figure 3) [95]. This FET-based biosensor provides high sensitivity and rapid diagnosis with the help of a small sample quantity. The sensing area of the FET biosensor was made with a 2D sheet of graphene NPs due to their excellent properties such as high carrier mobility, high electronic conductivity and a large surface area. Here, they have functionalized the graphene-based biosensor with SARS-CoV-2 spike antibody through a probe linker, 1-pyrenebutyric acid N-hydroxysuccinimide ester (PBASE). Clinical COVID-19 patient samples were evaluated using nasopharyngeal swab specimens, cultured virus samples and antigen proteins, and a positive response was observed within a few minutes with a limit of detection of 1 fg/mL. Also, this FET-based biosensor was able to distinguish the SARS-CoV-2 antigen proteins from those of MERS-CoV [95].

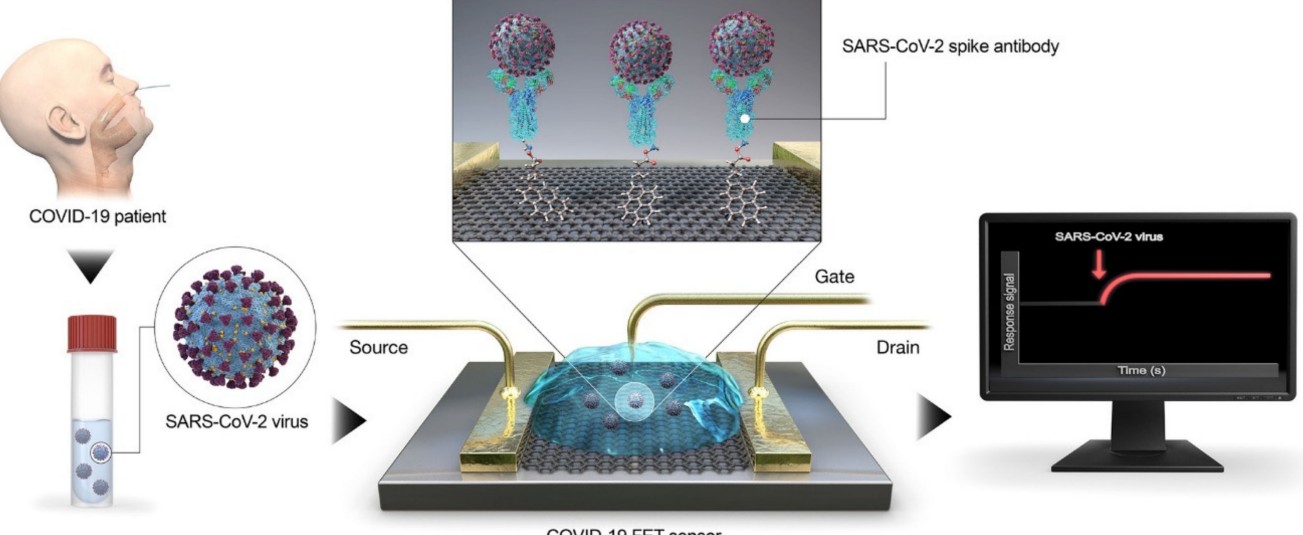

**Figure 3.** Schematic diagram of COVID-19 FET sensor operation procedure. Graphene as a sensing material is selected, and a SARS-CoV-2 spike antibody is conjugated onto the graphene sheet via 1-pyrenebutyric acid N-hydroxysuccinimide ester, which is an interfacing molecule as a probe linker. Adopted with permission from Ref. [95]. Copyright 2020, American Chemical Society.

Among the various inorganic NPs, magnetic nanoparticles (MNPs) are often utilized for nucleic acid extraction, purification, and detection [96]. Some of the benefits of MNPs for these applications include superparamagnetic behavior, low cost of preparation, quick isolation inside buffer solutions, and signal detection sensitivity [96,97]. Accurate detection necessitates efficient nucleic acid extraction and separation from samples, allowing for target purification [96]. For instance, a hybrid MNP system consisting of chitosan-coated ferrite MNPs integrated with graphene oxide has been used for the extraction of RNA to detect SARS-CoV-2 in patient samples [98]. Researchers can effectively reduce the risk of false negatives by extracting viral RNA using these methods prior to RT-PCR testing [98]. In order to detect SARS-CoV-2 early on by PCR, the RNA has to be converted to complementary DNA (cDNA) as only DNA can be copied or amplified. In one study, superparamagnetic nanoparticles (80 nm) were functionalized with an oligonucleotide probe to capture viral complementary DNA of SARS-CoVs [99]. These functionalized superparamagnetic nanoparticles have the potential to extract target cDNA from specimens

using a magnet. Then, PCR was used to increase the amount of isolated DNA, and it was evaluated using silica-coated fluorescent nanoparticles conjugated with a complimentary sequence [99]. Fluorescent signals produced by silica-coated fluorescent NPs are precisely proportional to the concentration of the target cDNA [99].

Quantum dots (QDs), also called "semiconductor nanomaterials", have also been used for COVID-19 diagnostics with diameters ranging from 1 to 10 nm [100]. QDs' unique optical properties have made them an excellent choice for use as a fluorescent label. Furthermore, by varying their size, their emission wavelength can be simply and precisely controlled [100]. QDs can be used as an imaging tool despite the use of microscopy, due to their superior fluorescence characteristics. For example, a smartphone-based quantum barcode serological test was developed for real-time monitoring of patients with SARS-CoV-2 at varied sampling times and infection severity levels in one research study by the Chan group [101]. In comparison to the data obtained from lateral flow assays (sensitivity of 34% and specificity of 100%), the acquired clinical sensitivity was around 90% and the corresponding specificity was 100% for the virus recorded by the smartphone-based quantum barcode serological test [101]. This smartphone-based device enables real-time monitoring of SARS-CoV-2 infection with varied sampling days based on the decrease in SARS-CoV-2 antibodies over time [101]. There are a few more examples reported for detection of COVID-19 using nanomaterials, as shown in Table 1.

**Table 1.** Nanoparticles used for diagnosis of COVID-19.

| Nanoparticles | Conjugates | Detection Technique | Target | Limit of Detection | Ref. |
|---|---|---|---|---|---|
| AuNP | Fluorine-doped tin oxide electrode (FTO) and nCOVID-monoclonal antibody | Immunosensor | SARS-CoV-2 virus | 90 fM | [102] |
| | IgM antibody | Lateral flow assay | SARS-CoV-2 virus | N/A | [103] |
| Lanthanide doped polystyrene NPs | Nucleocapsid phosphoprotein of SARS-CoV-2 | Lateral flow immunoassay (LFIA) | IgG in serum | N/A | [104] |
| Magnetic NPs (MNPs) | poly (amino ester) with carboxyl groups (PC) | RT-PCR | Viral RNA | 10 copies | [105] |
| Quantum dots (QDs) | RNA aptamer | Fluorescent sensor | SARS-CoV N protein | 0.1 pg mL$^{-1}$ | [106] |
| Reduced graphene oxide | Viral antigen | Electrochemical Biosensor | SARS-CoV-2 spike S1 protein | $1 \times 10^{-12}$ M $1 \times 10^{-15}$ M | [107] |
| Titanium Dioxide nanotubes | Spike Receptor Binding Domain (RBD) | Electrochemical sensor | SARS-CoV-2 | 0.7 Nm | [108] |

## 4. Therapeutic Strategies in COVID-19 Treatment

There are many therapeutic strategies being tested as potential targets in COVID-19 treatment. Several drugs used to treat other conditions have been suggested as possible candidates for inhibition of this virus. Some antiviral regimens that have been evaluated for SARS-CoV-2 contain a specific mechanism of action that involves the change of viral surface proteins responsible for virus binding and entry into the cell surface [109]. Although research into the efficacy and potency of these agents is ongoing, it is important to note that these antivirals have been used for specific targets and have the disadvantage of developing resistance due to viral mutation. The main limitations of antiviral drug candidates are the lack of specific targeting that results in host cell cytotoxicity.

*Nanoparticle-Based Therapeutic Applications*

Although many therapeutic strategies are in the pipeline, nanoparticles (NPs) represent a promising and potential tool against COVID-19. The versatility of nanoparticles makes

them tunable vectors for site-specific virus-targeting and drug-delivery carriers. Effective NP-based treatments can be created with care because they can be customized to prevent receptor binding and viral entry in cells, obstruct viral genetic material replication and proliferation, and work directly to inactivate viruses [110]. These nanomaterials may be combined with active antiviral drugs to improve the solubility and pharmacokinetic parameters and provide better therapeutic efficacy and fewer side effects [111]. A variety of nanomaterials are being used in theranostics for COVID-19, as shown in Scheme 1.

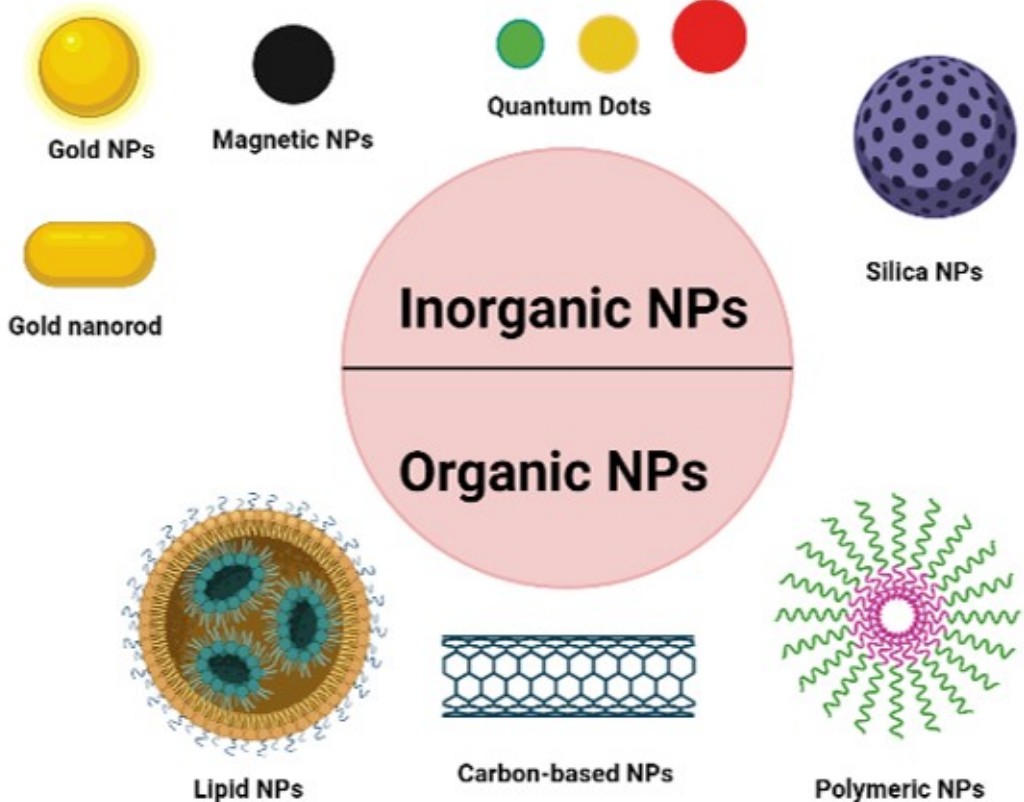

**Scheme 1.** Types of nanomaterials used for diagnosis and therapeutics of COVID-19.

Over the years the use of inorganic nanoparticles has created a potential opportunity for novel antiviral therapies, since metals may attack a broad spectrum of viruses and are less prone to develop resistance as compared to conventional antivirals [112]. An extensive review by Imani et al. described that this antiviral effect can be achieved by coating nanoparticles modified by different efficient biomolecular covers [109]. These functional coating techniques can inactivate the virus directly or identify and target the receptors on the host-cell surface and inhibit viral entry into cells. Among the metal nanomaterials, silver nanoparticles (AgNPs) have received enormous attention due to their antiviral [113], antibacterial, as well as optical, electrical, and thermal properties. Ryo and group reported the concentration dependent antiviral effect of AgNPs on SARS-CoV-2 virus [14]. Furthermore, their results indicated the AgNPs inhibited viral entry via disturbing viral integrity [14]. An outcome of one recent study suggested that AgNPs are potent against SARS-CoV-2 virus but showed higher cytotoxicity towards mammalian cells [14].

The excellent optical properties of AuNPs at the nano level make them ideal candidates for variety of applications such as diagnosis, therapy and imaging [114,115]. Additionally, the ease of surface functionalization using the strong covalent bond between gold atoms and thiol bond opens the door for conjugating biomolecules such as proteins, peptides, ligands, aptamers and antibodies. These strategies show promise and should be considered when researching antiviral therapies for SARS-CoV-2. The shape of a nanoparticle influences how it interacts with host cells and viruses. Gold nanorods have attracted significant

press coverage because of their optical, physicochemical and photothermal properties [116]. Extensive research has reported on utilizing gold nanorod's photothermal properties in the treatment of cancers, bacterial infections and viral infections [116]. Recently, Du et al. have developed a nanoplatform using gold/silver nanorods, (Au/AgNRs) which inhibited coronavirus multiplication using various mechanisms imparted by Au/AgNRs [117]. Their systematic study has proved that Au/AgNRs blocked viral entry into the cells and decreased the mitochondrial membrane potential. Furthermore, it was discovered by the authors that a large amount of reactive oxygen species was produced at the infection site, which could oxidize the silver shell on the surface of Au@AgNRs to release Ag+, achieving a long-term inhibitory effect on the entire virus replication cycle [117].

MNPs are widely employed in biomedicine, primarily as contrast agents in magnetic resonance imaging. The USFDA has approved various MNPs-based therapeutic products due to their biocompatibility [118]. The superparamagnetic behavior of tiny MNPs emerges in response to an external magnetic field [97]. Recent research suggests that SARS-CoV-2 interacts with hemoglobin in the erythrocyte or red blood cell progenitors, causing hemoglobin denaturation and iron metabolism dysregulation [119,120]. In several studies, individuals who did not survive had greater ferritin levels and lower hemoglobin levels [121]. As a result, these variations in iron levels may limit the use of these MNPs in therapeutic applications for COVID-19, as they may cause side effects [121,122].

Many organic NPs have already been approved by USFDA such as liposomes/lipid-based NPs to treat various bacterial, viral, and fungal infections and cancers [123]. This excellent nanoplatform has already entered in clinical studies and been approved by USFDA, with a well-known example being a liposomal version of doxorubicin (Doxil), which was approved by the USFDA in 1995 to treat the AIDS-related Kaposi sarcoma [124,125]. There has been extensive research published on the use of liposomes for drug delivery applications in the treatment of diseases [126]. In terms of COVID-19 application, recently Satta et al. engineered nano-liposome that can neutralize SARS-CoV-2 by human ACE2 (hACE2) effectively and could be used to treat COVID-19 [127]. LNPs are currently gaining attention as important elements of the COVID-19 mRNA vaccines because they are essential for successfully preserving and delivering mRNA to cells.

Several biodegradable polymer-based NPs have been approved by the USFDA to treat cancer and autoimmune disease and for use in bone regeneration [128]. Polymeric NPs are promising antiviral materials because they can be engineered to reach specific targets, prevent virus attachment to host cell receptors, and reduce undesirable interactions with the immune system, increasing the stability and pharmacokinetics of encapsulated drugs, the ability of controlled drug release and the ease of chemical modification [129]. Extensive research has been conducted on the use of polymeric nanoparticles to encapsulate hydrophilic, hydrophobic, and amphiphilic molecules in order to treat a variety of diseases [130]. The use of stimuli-responsive polymers (pH, light, enzyme) has been reported to improve controlled drug release. Hence polymeric NPs are a promising nanosystem for loading antiviral drugs and inhibiting viral infections, with improved controlled drug release, bioavailability and reduced side effects [131].

Researchers developed an approach to treat COVID-19 by saturating viral receptors with polymeric nanoparticles before the virus attaches, enters host cells, replicates, and infects. Nanoparticles conjugated to angiotensin converting enzyme (ACE2—Viral spike protein receptors) and CD147 proteins could saturate SARS-CoV-2 receptors, hence reducing their availability and blocking virus entry into host cells [132]. For example, Zhang et Al., have utilized nanosponges that were constructed by wrapping polymeric nanoparticle (NP) cores with natural cell membranes from target cells such as lung epithelial cells and macrophages while expressing the ACE2 and CD147 proteins, which are crucial for viral entry into cells [75]. Their systematic studies proved that, upon binding with the nanosponges, the coronaviruses were unable to infect their usual cellular targets [75]. This happens due to the binding of ACE2- and CD147-presenting NPs to viral proteins, which ultimately blocks viral ligands from adhering to host cell receptors [75].

Wang et al. reported a study in which, using ACE2-rich cells, they created membrane nanoparticles [45]. The selection of ACE2-rich cells was made because ACE2 is known to be the primary receptor for SARS-CoV-2 and to facilitate viral entry into host cells [45]. The ACE2-NPs contained an abundant amount of ACE2. Hence, through competitive inhibition, the ACE2-NPs bound to the virus and blocked the viral ligand from adhering to host cells in a dose-dependent manner and have potent capacity to block SARS-CoV-2 infection [45]. A brief compilation of COVID-19 therapeutic strategies is shown in Table 2.

**Table 2.** Therapeutic application of nanoparticles in COVID-19.

| Nanoparticles (NPs) | Conjugates | Virus | Mechanism of Action | Study Type | Key Conclusion for COVID-19 Treatment | Ref. |
|---|---|---|---|---|---|---|
| Mesopore silica nanoparticles | Niclosamide | SARS-CoV2 | Interference with spike protein of virus, hence hinder their adhesion to human cell receptor | In vitro In vivo | Inhibit SARS-CoV-2 Infectivity by targeting and therapeutic effect from Niclosamide | [133] |
| Iron oxide NPs | - | SARS-CoV2 | Interference with the virus binding and entry to the host cells. | Molecular Docking | Models were effective to be bonded with viral proteins by causing conformational changes and viral inactivation | [134] |
| Nano emulsion | Favipiravir | SARS-CoV2 | Interfere with the virion envelope composition or mask the virus (SARS-CoV-2) building | - | Hinders virus replication through inhibiting the SARS-COV-2 RNA replication | [135] |
| Lipid NPs | m-RNA 1273 | SARS-CoV-2 | Viral inactivation | In vitro In vivo | Significant reduction of viral load. | [136] |
| Silymarin–chitosan nanoparticles | Silymarin | SARS-CoV2 and ADV-5 | Blocking viral host receptor ACE2 | In vitro | Preventing viral attachment and its entry to the cells | [137] |
| Silver NPs | - | SARS-CoV2 | Interfere with the structural proteins of the virus by inhibiting their ability to bind with cell receptors | In vitro In vivo | Inhibitory effect on SARS-CoV-2 infectivity | [138] |

## 5. Role of Nanomaterials in Vaccine Development

Human lives have been seriously impacted by the COVID-19 pandemic, and desperate attempts are being made around the world to create safe and effective vaccines. Vaccination is the most effective solution to control and combat infectious disease. It has proved to be the most efficient and cost-effective prophylactic measure against many infectious diseases. Serious diseases such as smallpox, measles, mumps, rubella, diphtheria, tetanus, pertussis, polio, and yellow fever have been completely eradicated by use of vaccinations [139]. The key objectives of COVID-19 vaccine development are to prevent serious illness and to prevent the spread of the virus at the population level. It is widely agreed that the cellular defense against SARS-CoV-2 will mainly come from neutralizing antibodies, which primarily prevent the entry of the virus into cells by blocking the interaction between the SARS-CoV-2 spike protein receptor binding domain (RBD) and the ACE receptors on the target cell membrane. However, the role of T cells in the defensive immune response by clearing infected cells is emphasized by other researchers. Probably, the best strategy is to provide a combination of antibody and T cell responses for an effective vaccine. In addition to the immunological aspects, other important dosage and logistical considerations should also be taken into account in the ideal vaccine development: route of administration, ideally oral or intranasal and in a single dose, easy and quick to manufacture and scale up, and long-term stability at room temperature to allow storage and transport in non-developed

countries. Essentially, when developing a vaccine, it is mainly important to identify the antigen, the adjuvant, the manufacturing method and the delivery strategy.

In modern vaccine design, nanotechnology platforms provide great utility and have helped catalyze novel candidate vaccines at an unparalleled pace for clinical research. Nanoparticles can be delivered through the oral, intranasal, subcutaneous and intramuscular routes, providing a crucial advantage by overcoming tissue barriers, lymph nodes, mucosal and epithelial barriers (airway, nasal, gastrointestinal, etc.) [140,141]. Nanoparticles have the ability to enter cells, allowing them to deliver mRNA and DNA vaccines, triggering the cells to express antigens. Additionally, nanoparticles can specifically target immune cells for the direct delivery or antigens, as in subunit vaccines. Nanoparticles can be packed with a broad variety of antigenic variants (by chemical conjugation or physical trapping or encapsulation) and an appropriate antigenic display makes them a relevant alternative in vaccinology compared to traditional approaches [142,143]. The main advantage of vaccine nanocarriers is their nanosize, which is on the same scale as many biological systems, such as viruses (including SARS-CoV-2). Emerging nanotechnologies, such as mRNA vaccines delivered by lipid nanoparticles and viral vector vaccines, have already entered phase II and III clinical trials alongside inactivated vaccines. These nano-structured particles allow antigens to be properly exposed to the immune system, to be protected from known degradation enzymes such as protease and nuclease, or to be associated with other compounds with adjuvant activity [144].

### 5.1. Antigen Delivery via Nanoparticle Vaccine

Antigen functionalization within or on the surface of nanocarriers depends on several factors, including the physicochemical characteristics of the nanoparticles and antigens, the biological stability, and the target sites. The association of antigen with nanoparticles are of different types: physical adsorption, chemical conjugation, electrostatic interaction, and biomimetic coating. Physical adsorption or entrapment of antigens on nanoparticles is dependent on nanomaterials' surface charge and non-covalent hydrophobic interactions. Due to their large curvature radii, synthetic nanoparticles have elevated surface energy [145]. As a consequence, protein adsorption can occur spontaneously due to a combination of weak interactions that lead to the development of the protein corona. The adsorption method has been extensively applied to nanocarrier-associated antigens, including gold nanoparticles [146], carbon [147], silica [148,149], and organic polymers [150,151]. It is worth mentioning that the formation of protein corona is a dynamic process that can be highly affected by both nanomaterials and antigens of interest [152]. It was previously reported that antigens can undergo conformational changes once absorbed on nanoparticles' surface which might have impact on vaccine formulation [153].

Chemical conjugation of protein or peptide antigens via thiol and amine groups is widely used for the bioconjugation of nanoparticles. Thiol groups react strongly with gold surfaces due the intense interaction between the sulfur and the gold, forcing the sulfur atom to fill the free orbitals of the gold atom, forming a covalent bond [154]. Such a concept has been widely extended to the interaction of nucleic acid antigens with gold nanoparticles [155–157]. Amine groups, on the other hand, are present on both protein and peptide antigens that can be associated to nanocarriers by amide bonding, usually by carbodiimide crosslinker chemistry. Amine-containing antigen and nanoparticles bioconjugation commonly performed with 1-ethyl-3-(3-(dimethylamino)propyl)carbodi-imide (EDC) and N-hydroxysuccinimide (NHS) where NHS or its sulfonate (sulfo-NHS) is effectively linked to carboxyl groups to form NHS esters with the help of EDC [158]. The NHS esters then conjugate covalently to form an intermediate compound with the primary amines, which is then hydrolysed to the desired conjugate. Many nanoparticle vaccines have been prepared using the EDC/NHS chemistry conjugation method [159,160]. Electrostatic attraction between oppositely charged nanoparticles and antigens has been used to create nanoparticle vaccines. Typically, cationic nanocarriers are prepared for the association with anionic protein antigens or vice versa. Several nanoparticle vaccines have been prepared using

electrostatic attraction strategies between antigens and nanoparticles [161–163]. The encapsulation and entrapment of antigens within a nanocarrier is used to avoid their biological degradation. Liposome [164,165], polymeric nanoparticles [166–168], and gold nanoparticles [169,170] are ideal nanoplatforms for 4 encapsulating antigens and providing controlled biological release [171].

*5.2. Adjuvant Delivery via Nanoparticle Vaccine*

The role of adjuvants in vaccines is to stimulate the immune response to an antigen that is co-delivered. Adjuvants are incorporated as an individual agent in an antigen or as conjugate entities by chemical fusion directly with an antigen. More recent advances in adjuvant research have identified several pathogen-associated molecular patterns (PAMPs) as promising adjuvant candidates for the promotion of both humoral and cellular responses [172] when traditional adjuvants such as alum have been commonly used clinically to promote only humoral responses [173]. It is understood that several PAMPs, such as CpG-ODN and cyclic dinucleotides, as well as other toll-like receptors (TLR) and molecular agonists, such as imiquimod and resiquimod, induce strong immune responses [174]. Their efficacy, however, raises safety issues over the potential induction of systemic inflammation. In order to direct these immunological modulators to lymph nodes, increase their effective concentration and decrease their systemic reactogenicity, nanoparticle-based delivery thus offers a desirable strategy. For instance, several COVID-19 vaccines use adjuvants to increase the immune response. Novavax's adjuvant, Matrix-M, increases the immunogenicity of the vaccine by recruitment of APCs at the injection site, thus increasing the antigen presentation to T-cells in draining lymph nodes [175]. Additionally, Vaxine's adjuvant, called Advax, is a polysaccharide microparticle derived from polyfructofuranosyl-D-glucose, which boosts the intrinsic immunostimulatory nature of the antigen [176]. BioNTech/Pfizer and Moderna do not specifically state the use of an adjuvant in their vaccines, but RNA already comprises immunostimulatory properties and signals through pathogen recognition receptors [177]. There is also a possibility that the LNP carriers they use will confer adjuvant properties on their own. Since the discovery of liposomes the addition of dicetyl phosphate offered greater protection against diphtheria toxoid than unmodified liposomes, and the study of the use of lipids as adjuvants also increased considerably [178]. It is inferred from the above discussion that nanoparticles provide opportunities for antigen and adjuvant codelivery to target lymph nodes and APCs, and promising literature evidence indicates that antigen and adjuvant codelivery increases vaccine potency at lower doses and decreases side effects [179,180].

Development of vaccines normally takes decades. The production of vaccine candidates against COVID-19, however, has been greatly accelerated. The similarity between SARS-CoV-2 and other viruses (mainly SARS-CoV and MERS-CoV), together with prior knowledge of their defensive immune responses, has been a significant factor in the accelerated development of COVID-19 vaccines. Health agencies, such as the WHO, maintain an overview of the global trend of vaccines in development against COVID-19. As of June 2022, there are around 166 and 198 vaccines are in clinical and pre-clinical development respectively [181]. Some of the candidate vaccines, including nucleic acid (mRNA and DNA) vaccines, subunit vaccines, peptide-based vaccines, inactivated or live attenuated virus vaccines, and replicating or non-replicating virus-based vaccines, are available (as of June 2022) at various stages of clinical trials with their route of administration (ROA) such as intramuscular (IM) (Table 3).

**Table 3.** COVID-19 vaccine candidates in Phase 3 clinical trials [181]. ROA—Route of Administration; IM—Intramuscular.

| Vaccine Designer | Vaccine Platform | Vaccine Type | ROA | Clinical Trials Registry |
|---|---|---|---|---|
| Sinovac Research and Development Co., Ltd., Beijing, China | Formalin inactivating whole virus particles combined with an alum adjuvant | SARS-CoV-2 vaccine (inactivated) | IM | NCT04456595 Study Report NCT04508075 NCT04582344 NCT04617483 |
| Sinopharm + Wuhan Institute of Biological Products, Beijing, China | Inactivated SARS-CoV-2 vaccine (Vero cell) | Inactivated virus | IM | ChiCTR2000034780 ChiCTR2000039000 NCT04510207 NCT04612972 |
| Sinopharm + Beijing Institute of Biological Products, Beijing, China | Inactivated SARS-CoV-2 vaccine (Vero cell) | Inactivated virus | IM | ChiCTR2000034780 NCT04560881 NCT04510207 |
| Bharat Biotech International Limited, Hyderabad, India | Whole-Virion Inactivated SARS-CoV-2 Vaccine (BBV152) | Inactivated virus | IM | NCT04641481 CTRI/2020/11/028976 |
| AstraZeneca + University of Oxford, Cambridge, Oxford, England | ChAdOx1-S—(AZD1222) (Covishield) | Viral vector (non-replicating) | IM | ISRCTN89951424 NCT04516746 NCT04540393 NCT04536051 |
| CanSino Biological Inc./Beijing Institute of Biotechnology, Beijing, China | Recombinant novel coronavirus vaccine (Adenovirus type 5 vector) | Viral vector (non-replicating) | IM | NCT04526990 NCT04540419 |
| Gamaleya Research Institute; Health Ministry of the Russian Federation, Moscow, Russia | Gam-COVID-Vac Adeno-based (rAd26-S+rAd5-S) | Viral vector (non-replicating) | IM | NCT04530396 NCT04564716 NCT04642339 |
| Janssen Pharmaceuticals, Beerse, Belgium | Ad26.COV2.S | Viral vector (non-replicating) | IM | NCT04505722 NCT04614948 |
| Novavax, Gaithersburg, MD, USA | SARS-CoV-2 Rs/Matrix M1-Adjuvant (Full length recombinant SARS CoV-2 glycoprotein nanoparticle vaccine adjuvanted with Matrix M) | Protein subunit | IM | NCT04611802 EUCTR2020-004123-16-GB NCT04583995 |
| Anhui Zhifei Longcom Biopharmaceutical + Institute of Microbiology, Chinese Academy of Sciences, Anhui, China | Recombinant SARS-CoV-2 vaccine (CHO Cell) | Protein subunit | IM | ChiCTR2000040153 NCT04646590 |
| Moderna + National Institute of Allergy and Infectious Diseases (NIAID), Massachusetts, USA | Lipid nanoparticle mRNA vaccine (mRNA-1273) | mRNA vaccine | IM | NCT04470427 |
| BioNTech + Fosun Pharma; Jiangsu Provincial Center for Disease Prevention and Control + Pfizer, Mainz, Germany, Shanghai, China, Newyork, USA | Lipid nanoparticle mRNA vaccine (BNT162b2) | mRNA vaccine | IM | NCT04368728 |

On 11 December 2020, the USFDA authorized Pfizer and BioNtech's first COVID-19 nano vaccine for emergency use, followed 7 days later by the emergency use authorization of the Moderna COVID-19 nanovaccine which was a big milestone in the fight against this devastating pandemic that has impacted so many families in the United States and around the world. Now, both the Pfizer/BioNtech and Moderna vaccines have received full USFDA approval and are marketed as Comirnaty® and Spikevax, respectively. The rapid development of COVID-19 vaccines was possible because the genome and structural knowledge of SARS-CoV-2 was made available in record time [182–184]. In the following section, we look into traditional as well as next generation nanovaccine approaches for COVID-19.

*5.3. Traditional Vaccines*

Active immunization against viruses has typically been focused on the use of entire pathogens in a weakened or killed form by chemical or physical methods, resulting in clinically approved treatments. Traditional vaccines hold merit, and here we emphasize

how live attenuated, inactivated, and subunit vaccine development efforts make the reality of the SARS-CoV-2 vaccine tangible.

### 5.3.1. Inactivated Vaccines

Inactivated vaccines are made with whole viral particles, which are inactivated by heat or chemicals. This vaccine mainly induces specific humoral immune responses with antibodies capable to block virus entry into target cells. These vaccine formulations are incapable of replication and are safer than live attenuated vaccine. However, their inactivation results in reduced immunogenicity and the need for multiple dose regimens to provide long-lasting immunity. Additionally, these vaccine formulations also frequently require adjuvants due to immune senescence to immunize the aging population. Specifically, inactivated vaccines against SARS-CoV-2 are under Phase 3 clinical evaluation developed by Sinovac [185], Sinopharm [186], and Bharat Biotech International Ltd. [187].

### 5.3.2. Live Attenuated Vaccines

These vaccines are made up of the entire virus, but the infectivity has been reduced so that the virus can replicate and activate the immune response without triggering the disease. However, these vaccines pose risks, such as the possible reversion by mutation to a virulent form or recombination with infectious wild-type strains, and they can cause disease in immunocompromised individuals [188]. Codagenix and Serum Institute of India have developed a live attenuated virus vaccine based on viral codon optimized SARS-CoV-2, which are under preclinical stage [189]. A single dose of YF17D-vectored SARS-CoV-2 vaccine YF-S0 that induces robust immune responses in hamsters and mice that makes it a possible vaccine candidate for SARS-CoV-2 [190]. Another approach by deleting virulence genes and introducing attenuating mutations by reverse genetic methods was previously used for SARS-CoV and MERS-CoV vaccines.

### 5.3.3. Subunit Vaccines

The vaccine is developed from selected viral antigenic protein subunits, so it is called subunit vaccine. Therefore, relative to a full virus vaccine, this type of vaccine has a lower risk of adverse reactions. Subunit vaccine candidates comprise minimal structural components of SARS-CoV-2 that when administered with adjuvants for enhanced immunogenicity, can promote protective immune responses in the host. The S-protein of SARS-CoV-2 is used by most subunit COVID-19 vaccines, in whole or in separate fragments, in particular RBD. Novavax, whose vaccine is in a phase 3 clinical trial, is the leader among developers [175]. Clover Biopharmaceuticals, Vaxine, Medigen Vaccine Biologics, and Sichuan University are independently subunit vaccines undergoing phase I clinical trials [181]. Additionally, subunit vaccines comprise viral proteins that are introduced in synthetic nanomaterials, virus-like particle (VLP) and protein cages, which serve as adjuvants and/or delivery vehicles. For instance, influenza virus vaccine (Inflexal V) developed by Crucell is a liposomal formulation that includes haemagglutinin flu protein [191]. Hence, the use of nanomaterials in the formulation of a subunit vaccine may therefore have a great chance of developing a next generation vaccine.

### 5.4. Next-Generation Vaccines through Advances in Nanotechnology

Next-generation vaccines are produced using the viral protein gene sequence data. If the viral protein(s) required for infection defense and thus inclusion in the vaccine (i.e., the vaccine antigen) are identified, the availability of coding sequences for the viral protein(s) is sufficient to begin vaccine production rather than relying on the virus's ability to grow. The next significant element in vaccine design is the selection of carrier. In designing next-generation vaccines, the nanotechnology approach offers enormous benefits: higher vaccine thermostability, targeted delivery for optimum immune response, and biological stability. Nanoparticles have benefits as vaccine carriers because they imitate the structural characteristics of viruses. Additionally, nanocarriers provide stability and targeting of

these payloads to antigen-presenting cells (APCs). Through their innate adjuvant behavior, nanocarriers can synchronize both antigen and adjuvant deliveries to target immune cells. The benefits of these nanocarriers allow the development of novel vaccine technologies for the next generation. From the point of view of vaccine technology growth, this is an exciting time when biotechnologists and nanotechnologists are working together, and for the first time, nanotechnology-based vaccines have made a widespread clinical impact. These nanoplatforms are extremely adaptable and greatly accelerate the development of vaccines, which is evident from the fact that most current clinical trials of SARS-CoV-2 vaccines include a next-generation model [192].

### 5.4.1. Viral Vector Vaccines

Viral vector vaccines contain the attenuated recombinant virus (as a virus vector) that carries the SARS-CoV-2 genome. Viral vector vaccines can be either non-replicating or replicating. Replicating vector vaccines infect the cells in which the vaccine antigen is generated, as well as more infectious viral vectors capable of infecting new cells that will then produce the vaccine antigen. Initially, non-replicating vector vaccines enter cells and generate vaccine antigens, but new particles of the virus are not produced. Since viral vector vaccines result in the development of endogenous antigens, both humoral and cellular immune responses are stimulated.

One benefit of these viral vector-based vaccines is that a single dose can be enough for protection, as in the case of vesicular stomatitis virus-based Ervebo vaccine against Ebola virus [193]. These vaccines use an inactivated viral vector, such as a "modified vaccine virus Ankara" (MVA) or adenovirus to express viral proteins, so that proteins can be identified by the immune system to produce an immune response. Several vaccines, including one against MERS coronavirus, which is closely linked to SARS-CoV-2, have already been successfully produced using the MVA vector [194,195]. A clinical trial for this MERS vaccine has already been completed and clinical testing for SARS-CoV-2 is currently under way [195]. CanSino Biologics, a Chinese company, is developing a non-replicating version of adenovirus 5 (AdV5)-based vaccine that carries the genome for SARS-CoV-2 S protein [196]. It is undergoing phase 3 trials in some Asian countries and is currently licensed for used in the Chinese military [197]. Additionally, AstraZeneca and Janssen Pharmaceutica are using non-replicating viral vector vaccines that are undergoing Phase 3 clinical trials [181].

### 5.4.2. Nucleic Acid-Based Vaccines

A promising alternative to traditional vaccine methods is to directly deliver the genetic code for in situ production of viral proteins. Nucleic acid-based vaccines can be made up of DNA or mRNA and can be easily modified as new viruses appear, which is why they were among the very first COVID-19 vaccines to reach clinical trials. Although these systems are attractive in terms of safety, speed, stability and scalability, as seen previously with other emerging technologies, they bring a significantly greater risk of failure in clinical development [197]. The specific benefit of these vaccines is that in addition to antibodies and $CD4^+$ T cell responses, the DNA or RNA vaccine causes $CD8^+$ cytotoxic T-cell responses, which play a key role in the eradication of viruses. Though DNA vaccines provide higher stability over mRNA vaccines, mRNA is non-integrating and therefore does not pose a risk of insertional mutagenesis. In addition, the half-life, stability, and immunogenicity of mRNA can be adjusted through modifications in carrier design [198].

Nanotechnology-based strategies offer solutions to the delivery challenge by transporting the vaccines to appropriate cell populations and subcellular locations [199]. Although synthetic nanocarriers, including cationic liposomes, polymeric and lipid nanoparticles, have been used to transport DNA vaccines across cell membranes, targeted formulations could further enhance the nuclear translocation of plasmid DNA. Furthermore, nanotechnology platforms, including cationic nanoemulsions, liposomes, dendrimers or polysaccharide particles, have been used to improve the stability and distribution of

mRNA-based vaccines [198,199]. Two nanoparticle-based vaccines, BNT162b2, developed by Pfizer/BioNTech [200], and mRNA-1273, developed by Moderna, claimed to be 95% effective [201], which represented a big step forward in the battle against the COVID-19 pandemic. Moderna and Pfizer/BioNTech utilize mRNA vaccines in which antigens are encapsulated within lipid nanoparticles.

As of July 2022 there are four vaccines approved by the USFDA or authorized for emergency use: Pfizer/BioNTech's mRNA-based vaccine (Comirnaty®), Moderna's vaccine (Spikevax), the Novavax COVID-19 vaccine, and the Johnson & Johnson/Janssen COVID-19 vaccine. Of these vaccines, two (Pfizer/BioNTech and Moderna) use lipid nanoparticles (LNPs) as a carrier that protects the mRNA. Naked mRNAs are susceptible to extracellular RNase and nuclease degradation, which is why it is important to formulate their delivery vehicle. LNPs are one of the most promising vaccine platforms, because they are highly effective in encapsulating DNA or RNA-based immunogens or antibodies. LNPs are virus-sized nanoparticles that are synthesized by the self-assembly of an ionizable cationic lipid. As shown by several studies, they have the ability to deliver mRNA efficiently into the cytoplasm [202]. Sustained-release kinetics of mRNA expression, and therefore protein translation, can be accomplished by preferring intramuscular and intradermal routes, providing high antibody titers and immune responses to both B cells and T cells.

The active ingredient of the Pfizer/BioNTech vaccine is mRNA that encodes for the receptor-binding domain (RBD) subunit of the SARS-CoV-2 S protein, which is responsible for initial attachment to the host cell through the ACE2 receptor [203,204]. The mRNA is modified with a single nucleoside incorporation of 1-methylpseudouridine, which not only decreases the immunogenicity of mRNA in vivo but also enhances its translation [205]. The mRNA is encapsulated within LNPs, as shown in Scheme 2, which are composed of four lipids: (4-hydroxybutyl)azanediyl)bis(hexane-6,1-diyl)bis(2-hexyldecanoate), which is the primary ingredient, and an ionizable cationic lipid, 2-[(polyethylene glycol)-2000]-N,N-ditetradecylacetamide (ALC-0159), 1,2-distearoyl-snglycero-3-phosphocholine (DPSC), and cholesterol, which provides structural integrity and increase the stability of lipid nanoparticles [206,207]. The presence of large PEG groups in lipids makes the LNPs' stealth, which shields the surface from aggregation, opsonization, phagocytosis, reducing immunogenicity and prolonging systemic circulation time. Ionizable lipids have positive charges that attach to the negatively charged backbone of mRNA, and phospholipid and cholesterol molecules contribute to the formation of the particle. Pfizer vaccines also contain four salts: potassium chloride, monobasic potassium phosphate, sodium chloride, and basic sodium phosphate dihydrate that maintain the pH of vaccine close to the individual's body pH [206]. It also includes sucrose, which serves as a cryoprotectant to secure and prevent the nanoparticles from sticking together when they are frozen [206]. The Pfizer vaccine is administered via intramuscular injections. Similar to Pfizer/BioNTech, Moderna has also developed mRNA vaccine by encapsulating mRNA within the LNPs that composed of four lipids. These vaccines are a significant achievement for molecular medicine, biotechnology and nanomedicine. They are a success for all those scientists who have worked to optimize the nano formulation for efficient packaging and safe delivery of genetic material.

### 5.4.3. Peptide-Based Vaccines

Peptide-based vaccines can be easily engineered, readily tested, and quickly produced [208]. The formulation of peptide-based vaccines consists of peptides plus adjuvant mixtures or peptides and adjuvant delivered by an appropriate nanocarrier. Several peptide-based and peptide nanoparticles-based vaccines are in clinical research and production for chronic diseases and cancer [209,210]. Peptide-based vaccines are also being developed against COVID-19. CoVac-1 is a peptide-based vaccine currently under Phase 1 clinical trial. CoVac-1, a multi peptide-based vaccine candidate, was developed by the Walz group, with the goal of inducing potent SARS-CoV-2 T cell immunity to fight against COVID-19 [211].

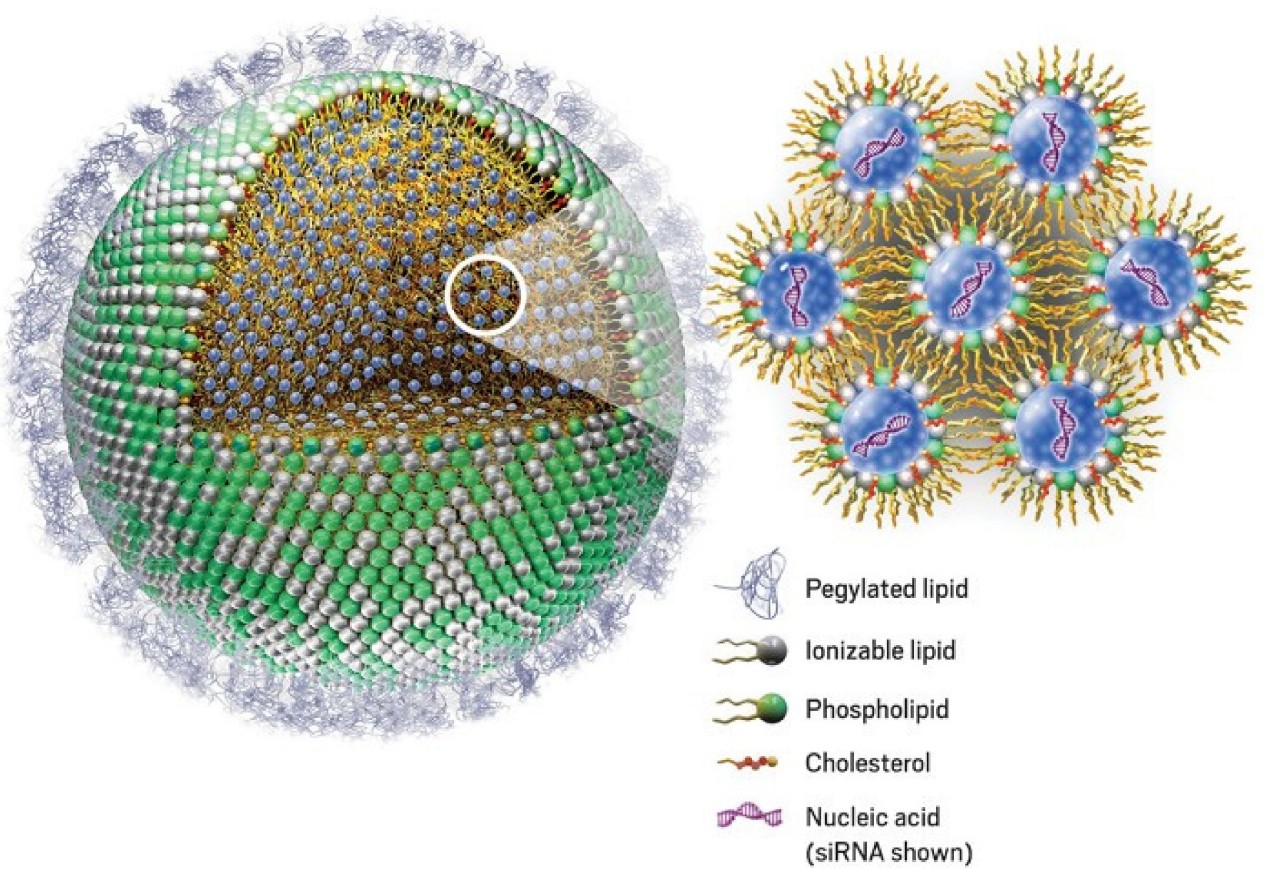

**Scheme 2.** A schematic of lipid nanoparticle (LNP) contains hundreds of small interfering RNA (siRNA) molecules, each surrounded by ionizable lipids, phospholipids, and cholesterol.

## 6. Conclusions

The severe nature of the COVID-19 global pandemic highlights the significance of new and innovative nanotherapeutic approaches as a means of limiting the spread of the deadly disease. Nanomaterial-based therapeutic regimes have emerged as effective treatment options in combating COVID-19 on account of their ability to deal with viral diseases in an effective manner and also addressing the limitations associated with traditional antiviral strategies. In this review, we discussed the advanced strategies involving nanomaterials for the rapid diagnosis, prevention, and effective treatment regimens associated to COVID-19. Herein, we have summarized a significant advance in the development of nanotechnology-based diagnostic nanotherapeutics including nanovaccines and nanoparticle-based therapeutics, single and combinational antiviral therapies against COVID-19. The exceptional properties of nanomaterials, including large surface-area-to-volume ratio, tunable optical properties, controllable sizes and excellent binding ability with biomolecules, play an important role in a broad range of applications. Owing to their unique physical and chemical properties, these nanomaterials-based approaches have attracted broad interest in the treatment of viral infections. There are still some challenges such as fibrosis and oxidative stress as well as genotoxicity that need to be explored before harnessing these nano-based therapies in clinical use. Although these intrinsic factors have not yet been fully resolved, based on the extensive literature review presented here, these nanotechnology-based systems are essential for the safe and effective implementation of nanotherapies to manage COVID-19 infection and to control the emergence of new viral infections more comprehensively and efficiently.

**Author Contributions:** Resources, U.P.; Writing—original draft preparation, writing, editing and review, K.R.; Writing and review, E.C.H.; Writing—review and editing, N.S.; Writing. All authors have read and agreed to the published version of the manuscript.

**Funding:** This research received no external funding.

**Data Availability Statement:** This study did not report any data.

**Conflicts of Interest:** The authors declare no conflict of interest.

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
