# Peer review of "Role of Nanomaterials in COVID-19 Prevention, Diagnostics, Therapeutics, and Vaccine Development"

_jnt, doi:10.3390/jnt3040011_

Round 1

Reviewer 1 Report

Very well and clearly structured article, well documented

Reviewer 2 Report

The review is often repetitive and difficult to read. Very general information are given in each section for each application and material. The characteristic of the nanomaterials are poorly described, and often their general antimicrobic performances are listed, and not their actual use in the fight to  the pandemic or theis  specific activity against SARS-COV-2.

The results is a work that contains few information for an expert in the filed of nanomaterials. Although the number  of the papers cited is adequate, the results of these works are just reported in very  general terms. Finally, the addition of scheme and tables summarizing the type and function of  NPs for each application field would be beneficial for thee reader.

Reviewer 3 Report

Patel et al. contributed a review, which summarizes the „Role of Nanomaterials in COVID-19 Prevention, Diagnostics, Therapeutics, and Vaccine Development“(title of the manuscript). Hence, the topic addressed by this timely review has a very high relevance (fighting coronaviruses) and the use of nanomaterials to reach this goal is an emergent field in science and witnessed tremendous improvements over the past years. From this point of view, the contributed review makes, in principle, an important contribution to the field that will be highly appreciated by the readers. However, the manuscript has severe shortcomings with respect to its structure, the provided information, and its presentation (which will be given in detail below), which require a MAJOR REVISION of the manuscript before any recommendation can be given. In fact, the manuscript contains many errors, which make me believe that it was not well checked by the authors before its submission and that part of this quality management was outsourced (to the reviewers), which (if true) would be inacceptable. Furthermore, I’m not fully convinced that the review fully matches to the scope of the Journal of Nanotheranostics, as I’m not aware of any SARS-CoV-2 related nanomaterial, which can be simultaneously used for diagnostic and therapeutic purposes (which is to my knowledge the definition of theranostics).

Nevertheless, in the following paragraphs, I will highlight some shortcoming, which have to be addressed by the authors.

1. General Remarks

1.1. Line 14 “We critically evaluate …”: Maybe I missed it, but I did not see any critical evaluation of the content. For example, the authors included a large body of studies, which introduce materials that have a promising performance in fighting SARS-CoV-2, but in my opinion, only two strategies (lateral flow assays and mRNA-containing lipid nanoparticles) have clinical relevance. A critical evaluation could be, why so many of these materials do not end up in the management of COVID19 (i.e., what is the shortcoming which hinders their clinical application?).

1.2. Line 16 “This review is highly insightful and beneficial in offering direction for creating products based on nanomaterials to combat COVID-19.”: The reader should decide, if she/he finds this review is highly insightful and beneficial.

1.3. The authors should check, if all information are still up to date. Many online resources were accessed in December 2021 (I hope), which is at least 8 months ago and quite some time in this rapidly developing field. I guess that the review was written over a longer period of time, so that not all information are still up to date. Compare for example line 72 “…both the BioNTech/Pfizer and Moderna vaccines, which were given USFDA approval, …” with line 466 “…Moderna close to obtaining approval by the USFDA.”, in which the latter sentence is definitely outdated.

2. Structure of the review

2.1. Section 2: To me it appears that the header “Role of Nanomaterials for Prevention of COVID-19” is too broadly worded, as prevention of COVID would also include diagnostics and vaccination (which will be covered in section 3 and 5), while section 2 rather focusses on nanomaterials that can be used for sterilization or sequestration of SARS-CoV-2.

2.2. The content of section 2 is not well condensed and some aspects appear to be redundant. It is full of sentences like “… have previously been proven to be effective against many different viruses [53-61], including coronaviruses [62]. These results indicate that AgNPs may be useful in the prevention of SARS-CoV-2.” (lines 98+99), “… copper-based nanomaterials to be effective against poliovirus [63] and other coronaviruses, such as SARS [64] and HuCoV-229E [65], indicating that copper nanomaterials may be useful in preventing SARS-CoV-2 transmission and infection.” (lines 105-107), “…that copper-based nanomaterials could provide a useful strategy for SARS-116 CoV-2 prevention…” (line 116) and so on.

2.3a. Section 4 is supposed summarize the “Role of Nanomaterials in Therapeutics of COVID-19”, which is a remarkable header, as I’m not aware of any nanomaterial currently used in the treatment of COVID-19. The authors mention here “conventional” drugs (such as antivirals), which I would not regard to be nanomaterials in general.

2.3b. Furthermore, inorganic and organic nanoparticles are mentioned, which has a notable overlap with the sections 2 and 5. Such overlaps should be avoided.

2.3c. Most of the examples of section 4 refer to other viruses and not to SARS-CoV-2, which is (according to the title of this section and of the review) off topic.

2.4. Section 5: The headings of this section do not always make sense. For example, the topic “Traditional Vaccines” has the same level as “Inactivated Vaccines” etc., although the latter is a subtopic of the former.

2.5. Section 5, line 603ff “Role of nanomaterials in vaccine development”. I don’t fully get the relevance of this part and the following parts of section 5 with respect of the scope of the review. These paragraphs are either quite general (i.e., cover a variety of different viruses excluding SARS-CoV-2 or topics such as thiol chemistry) or are redundant with respect to previous sections. For example, lines 655 to 695 cover (again) the lipid nanoparticles of Biontec/Pfizer and Moderna, so that some key information on these nanoparticles are given for the third time. I appreciate that new information are given by the paragraphs on adjuvants, but besides this, Section 5 needs to be restructured to avoid redundancies within the section and across the review.

3. Wrong or imprecise information

3.1. Line 203 “RT-PCR for SARS-CoV-2 detection is based on the determination of immunoglobulin (Ig) G and IgM antibody levels which starts after 5–7 days of infection, by using respiratory secretions.” This is utterly wrong. RT-PCR detects RNA by first translating the RNA to DNA (RT) and then by amplifying the DNA (PCR).

3.2. lateral flow immunoassays, line 222 “… using a little amount if patient blood … ” and line 230 “The blood or serum sample …”: These statements are in general true, but do not reflect the fact that in the majority of current applications swaps from nose or throat are used (and not blood samples).

3.3. Line 295 “QDs are now the most common imaging for sensing due to their superior capabilities.”: Besides the grammar issues, I wonder based on what criteria this assessment was made? To my knowledge, SARS-CoV-2 is typically visualized in cryo-EM and TEM either without labels or using immunogold or in fluorescence microscopy using dyes. The number of publications using these approaches should outnumber the number of publications using QDs by far.

3.4. Line 535 “Next generation vaccines”: The authors should give their definition of a next generation vaccine, i.e., based on which criteria does one distinguish between next generation and traditional vaccines? Furthermore, in contrast to this review, viral vectors are often regarded to be a next generation vaccine, while subunit vaccines are often listed together with other traditional vaccines (see e.g. van Riel, D., & de Wit, E. (2020). Next-generation vaccine platforms for COVID-19. Nature materials, 19(8), 810-812.).

3.5. Line 578 “Moderna and Pfizer/BioNTech utilize mRNA vaccines in which antigens encapsulated within the lipid nanoparticles.” Besides the grammar issues, this statement is wrong as mRNA vaccines do not encapsulate antigens in the lipid nanoparticles.

3.6. Line 614 “Nanoparticles have the ability to enter cells to trigger mRNA and DNA vaccines to express antigens …” The authors should word more clearly here. Do they really refer to nanoparticles and nucleic acid vaccines or is this the same entity? Furthermore, vaccines will not express antigens; this will be done by the cell, right?

4. Questions to the content

4.1. Line 180 “These membranes were shown to be effective in filtering particles with sizes below 300 nm [21], meaning they would be effective for preventing the spread of SARS-CoV-2.” This is worded in a misleading way, as an upper limit is given (sizes below 300 nm), while porous membranes should filter all particles, which exceed a lower size limit, right? What is this limit it in this case?

4.2. Line 194 “The biocompatibility of these polymers also makes them well suited to the treatment of COVID-19 and the prevention of SARS-CoV-2 through vaccination.” How can polymers be used for prevention of SARS-CoV-2 through vaccination? I’m not aware of vaccination strategies based on polymers.

4.3. Line 208 “Hence, they have developed … ”: Who is “they”?

4.4. Figure 1: The authors missed to indicate, from which publication this figure was taken.

4.5 Line 286 “… extract target cDNA from specimens … ”: Define cDNA and explain, why it is present in infections by a RNA virus.

4.6. Line 287 “… it was evaluated using silica-coated fluorescent nanoparticles conjugated with complimentary sequence.” What does “it” refer to and specify to what entity the sequence was complementary (and not complimentary).

4.7. Line 297 “…real-time monitoring of patients with SARS-CoV-2 at varied sampling times …”: What is the meaning of varied here (in the context of real-time monitoring)?

4.8. Line 322 “Carefully designed NP-based therapeutics can be an effective solution since they can be tailored to block receptor binding and viral entry in cell, hinder the viral genetic material replication and proliferation, act directly toward virus inactivation.” Is there any FDA-approved NP-based therapeutic available to date (beyond lipid nanoparticles)? The sentence reads as if NP-based therapeutics are already established for the applications mentioned in the sentence.

4.9. Table 2 “Key conclusion for COVID-19 treatment”: As viruses can behave very differently, how can findings from non-related viruses be translated into a conclusion for COVID-19 treatment?

4.10. Table 2 Entry of “Lipid NPs”: Here, it doesn’t make sense to indicate a protein (SARS-CoV-2 full-length S-protein) as target virus.

4.11. Line 426 “Nanoparticles were conjugated to angiotensin converting enzyme (ACE-2 – Viral spike protein receptors) and CD147 proteins, could saturates SARS-COV-2 receptors, hence reducing their availability and blocking virus entry into host cells.” Besides the problems with grammar, how can ACE2-presenting nanoparticles reduce the availability of ACE2?

4.12. Line 498 “These vaccines are live, made up of the entire virus, …” This is a bit misleading, as there is a debate if viruses are actually living entities. The authors probably want to express that here the viruses have not been inactivated but their infectivity has been reduced (e.g., using genetic approaches).

4.13. Line 550 “… this is an exciting time when biotechnologists and nanotechnologists are working together and are poised for the first time to have a clinical impact.” This implies that biotechnologists and nanotechnologists have not yet made clinical impact, which I would not agree with (counterexample: PCR).

4.14. Line 552 “This makes these hybrid platforms …”: Clarify the meaning of hybrid here. In which respect are these platforms hybrid?

4.15. Line 553 “… which is evident from the fact that most current clinical trials of SARS-CoV-2 vaccines include a next-generation model.” The authors should provide a reference to support this claim.

4.16. Line 597 “Peptide-based vaccines”: Is there any example for a peptide-based vaccine against SARS-CoV-2? If not, this should be stated explicitly in this section.

4.17. Line 606 “…developing the COVID-19 vaccine.”: To which one do the authors refer here? There are several available.

4.18. Line 610 “As many biological systems such as viruses (including SARS-CoV-2) and proteins are also nanosized, the main advantages of vaccine nanocarriers are their nanosize.” This is a tautology.

4.19 Conclusions, line 730 “Nanotechnologies have been emerged as a new and mighty era of effective treatment options in combating COVID-19.” This deserves justification by the authors, as the clinically relevant strategies presented here (lateral flow assays and lipid nanoparticles) are actually quite old. To my knowledge, mRNA delivery by lipid nanoparticles was demonstrated already decades ago, but I agree that the first FDA approval was given to the SARS-CoV-2 vaccines based on this concept.

4.20. Conclusions, line 735 “… nanovaccines …”: Are there also vaccines that are not nm-sized entities, i.e., what is the difference between a vaccine and a nanovaccine?

4.21. Conclusion, line 736 “Nanoparticles based drug delivery methods showed significant potential applications by its unique physiological characteristics.”: Besides the issues with grammar, I’m not sure if physiological characteristics of nanoparticles were covered in this review.

4.22. Conclusion, line 737 “Due to their tendency to overcome the drug resistance …”: Maybe I missed it, but where was this discussed in the review? As traditional vaccines are here included in the field of nanoparticle-based therapeutics, I actually doubt that this reasoning is correct.

4.23. Conclusion, line 739 “There are still some important challenges such as fibrosis, oxidative stress as well as genotoxicity that clearly needs to be explored in more in-depth understanding before harnessing the nano-based therapies in clinical use.”: This sentence also has to be revised (due to issues with syntax and grammar).

5. Presentation

Furthermore, the manuscript has a significant amount of typos and grammar mistakes. The following list just gives some examples, the density of which demonstrates that the manuscript was not properly checked before submission.

- Line 231 “… the antibodies start interacting with the antigen bound AuNPs and dragged through the chromatographic strip by capillary action, resulting corresponding lines”.

- Line 240 “… at the same time with a within 15 minutes …”.

- Line 264 “… through a prove linker, …”. I don’t get the meaning of prove in this context.

- Line 279 “For instance, hybrid MNPs system consisting ...”

- Line 283 “In one study, superparamagnetic nanoparticles (80 nm) conjugated with a probe that is complementary to the target sequence SARS-CoVs were used.” This sentence has some issues with the syntax and currently makes no sense, as a sequence cannot be complementary to a virus (only to nucleic acids).

- Line 301 “This newly developed device and assay enable real-time monitoring …”

- Table 1, 5th column header “Limit of detection with (time)”.

- Line 310 “Some antiviral regimens have been evaluated to contain SARS-CoV-2, have their specific mechanism of action that involves the change of viral surface proteins responsible for binding and entry of virus into cell surface.” This sentence has severe issues with the syntax.

- Line 312 “Although the investigation … are still in progress …”.

- Lines 328 – 330 give just a list of terms instead of formulating a correct sentence.

- Line 377 “Many Organic NPs has already …”.

- Line 385 “Liposomes or a earlier version of lipid nanoparticles has gained much attention due to their tremendous properties such as offers stability, encapsulate hydrophilic or hydrophobic drugs, biodegradable, nontoxic, ease chemical modifications for targeted applications.” has also major problems with the syntax.

- Line 392 “There is an extensive study has been reported of using …”.

- Line 415 “… of using polymeric nanoparticles to encapsulates …”.

- Line 424 “Researchers have developed an approach to treat COVID-19 is to saturate viral receptors with polymeric nanoparticles before virus could attach followed by entry into host cells, replicates and infect.”

- Line 430 “… there are multiple research reported promising outcomes in viral inactivation …”

- Line 488 “… which inactivated by heat or chemicals.”

- Line 489 “This vaccine mainly inducing specific humoral immune responses …”

- Line 513 “Phage 3 clinical trials.”

- Line 535 “Nanomaterials and viruses exist within the same length scale (20-200 nm in size) this is what makes nanotechnology approaches to next generation vaccine development and immune-engineering so powerful”.
